# Reconfiguring crystal and electronic structures of MoS$_2$ by substitutional doping

Joonki Suh [1,11], Teck Leong Tan[2], Weijie Zhao[3], Joonsuk Park[4], Der-Yuh Lin[5], Tae-Eon Park[6], Jonghwan Kim[7,12], Chenhao Jin[7], Nihit Saigal[8], Sandip Ghosh [8], Zicong Marvin Wong[2,9], Yabin Chen[1], Feng Wang[7,10], Wladyslaw Walukiewicz[1,10], Goki Eda [3] & Junqiao Wu [1,10]

Doping of traditional semiconductors has enabled technological applications in modern electronics by tailoring their chemical, optical and electronic properties. However, substitutional doping in two-dimensional semiconductors is at a comparatively early stage, and the resultant effects are less explored. In this work, we report unusual effects of degenerate doping with Nb on structural, electronic and optical characteristics of MoS$_2$ crystals. The doping readily induces a structural transformation from naturally occurring 2H stacking to 3R stacking. Electronically, a strong interaction of the Nb impurity states with the host valence bands drastically and nonlinearly modifies the electronic band structure with the valence band maximum of multilayer MoS$_2$ at the $\Gamma$ point pushed upward by hybridization with the Nb states. When thinned down to monolayers, in stark contrast, such significant nonlinear effect vanishes, instead resulting in strong and broadband photoluminescence via the formation of exciton complexes tightly bound to neutral acceptors.

[1] Department of Materials Science and Engineering, University of California, Berkeley, CA 94720, USA. [2] Institute of High Performance Computing, Agency for Science, Technology and Research, Singapore 138632, Singapore. [3] Department of Physics, National University of Singapore, 2 Science Drive 3, Singapore 117551, Singapore. [4] Department of Materials Science and Engineering, Stanford University, Stanford, CA 94305, USA. [5] Department of Electronics Engineering, National Changhua University of Education, Changhua 50007, Taiwan. [6] Center for Spintronics, Korea Institute of Science and Technology, Seoul 02792, Korea. [7] Department of Physics, University of California, Berkeley, CA 94720, USA. [8] Department of Condensed Matter Physics and Materials Science, Tata Institute of Fundamental Research, Mumbai 400005, India. [9] Department of Chemistry, National University of Singapore, 3 Science Drive 3, Singapore 117543, Singapore. [10] Materials Sciences Division, Lawrence Berkeley National Laboratory, Berkeley, CA 94720, USA. [11]Present address: Department of Chemistry, University of Chicago, Chicago, IL 60637, USA. [12]Present address: Department of Materials Science and Engineering, Pohang University of Science and Technology, Pohang 790-784, Korea. Correspondence and requests for materials should be addressed to J.S. (email: joonki@uchicago.edu) or to J.W. (email: wuj@berkeley.edu)

Substitutional doping of bulk semiconductors, the atomistic substitution with non-isoelectronic impurities, allows to define the type of majority charge carriers and modulate their concentrations to a wide extent, such that they can electrically functionalize as the key component in electronic and optoelectronic devices. For the emerging two-dimensional (2D) or layered semiconductors such as transition metal dichalcogenides, $MX_2$ (M = transition metal and X = chalcogen), substitutional doping is highly desired to overcome their natively unipolar conduction[1] and substantial contact resistance[2], despite recent efforts based on surface molecular doping[3,4] and phase transition[5,6]. In this context, substitutional doping in $MX_2$ has been recently experimentally realized, mainly by replacing the host cation M with other transition metal elements[7–9], often leading to degenerate doping levels of free carriers or even switch in conduction type that was hardly achieved by the other doping techniques[3]. Also, recent theoretical studies report doping-induced modifications in magnetic[10,11] and catalytic[12] properties of $MX_2$.

The atomic $d$-orbitals in transition metal dopants can exhibit different extents of localization[13]. Namely, they are originally spatially localized with a constant energy level relative to the vacuum level, but these discrete levels may interact with each other to gain dispersion as their wavefunctions overlap, particularly at high doping concentrations. It ultimately results in spatial delocalization of the wavefunctions. Consequently, impurity-related sub-bands may emerge near the band edge of the host, and at high densities they may eventually hybridize with the host bands, causing bandgap modification and redistribution of the density of states (DOS)[14,15]. As expected from the nearly linear, virtual crystal approximation widely adopted for semiconductor alloys[16], however, such band restructuring effects often require considerable concentrations of dopants exceeding a few atomic percent to reach the alloying level, with exceptions identified in the so-called highly mismatched alloys[15]. Such dopants-induced band restructuring effects are believed to become stronger at reduced dimensionalities. Heavy doping may also lead to exotic behavior in reduced dimensionalities. For instance, quantum-confined Urbach tail[17] and dynamic surface exciton quenching[18] were observed in heavily doped zero-dimensional (0D) nanocrystals and one-dimensional (1D) carbon nanotubes, respectively. Yet, delocalization of dopant wavefunction and the resultant band restructuring in 2D semiconductors are still experimentally unresolved thus far, despite the fact that quite unique electronic and optical characteristics[19–21] were observed in heavily doped 2D materials.

Among the suggested cation dopants substituting molybdenum (Mo) in molybdenum disulfide ($MoS_2$), niobium (Nb) has one less $d$-electron, and is of particular importance from the thermodynamics point of view: it is predicted to have a negative formation energy[22]. In addition, Nb doping in $MoS_2$ was suggested to induce considerable degree of charge delocalization as studied in electron paramagnetic resonance experiments[23] and first-principles calculations[22], as opposed to the case of native point defects (e.g., sulfur vacancies) which only induce localized and nondispersive (flat-energy level) states deep inside the bandgap[24].

In this work, we report modifications in both crystal structure and electronic structure caused by Nb dopants in monolayer to bulk $MoS_2$. We show doping-induced structural conversion from 2H to 3R stacking in the layered structure, accompanied by a renormalization of the valence band structure. In the multilayer or bulk, the wavefunctions of Nb dopants hybridize with the host valence band at the center of the Brillouin zone ($\Gamma_V$), thereby dramatically reducing the indirect bandgap. In stark contrast, such electronic restructuring is greatly suppressed in the ultimate 2D limit where $\Gamma_V$ of $MoS_2$ natively moves down, separating out the Nb states as isolated impurity states, i.e., ionization energy of the impurity states becomes greater due to intrinsically larger bandgap and reduced screening, eventually causing the formation of impurity bound excitons.

## Results

**Crystal structure of Nb-doped $MoS_2$.** Both Nb-free and Nb-doped $MoS_2$ single crystals, $Nb_xMo_{1-x}S_2$ ($x = 0$–0.01), were synthesized using the chemical vapor transport methods (see Methods). For the Nb-doped $MoS_2$ ($MoS_2$:Nb), three representative doping concentrations were prepared to reach the degenerate doping: 0.1, 0.5, and 1% (where the % is defined as the atomic percent of Nb with respect to the density of host Mo sites). In our previous study, X-ray absorption and structural analysis have been employed to verify the substitutionality of the Nb doping as well as uniform distribution of dopants with no phase separation[7]. In addition, Hall-effect technique confirms that free hole concentrations are consistent with the synthesis condition (Supplementary Figure 1). The polymorphs of the synthesized bulk crystals were determined by convergent beam electron diffraction (CBED), and typical images are presented in Fig. 1c. With an obvious difference in symmetries between the $MoS_2$ and $MoS_2$:Nb, the CBED patterns confirm the 2H and 3R stacking for the Nb-free and Nb-doped $MoS_2$, respectively. The 2H and 3R are two different types of stacking order of $MX_2$ in the trigonal prismatic coordination, described by the space groups of $P6_3/mmc$ and $R3m$ (Fig. 1a), respectively. Unlike the 2H phase, owing to the non-centrosymmetric crystal symmetry and the resultant pseudo-spin polarization in the valence bands, the 3R phase recently attracts great interests for study of valleytronics[25]. While the 2H phase is found more frequently for semiconducting $MX_2$, the 3R phase is thermodynamically also stable thanks to the nearly negligible difference between their ground-state total energies, and is observed in most of Nb-doped samples we tested (38 out of 41 crystals; see additional CBED images in Supplementary Figure 2 and Note 1). A triangular shape of screw dislocation spiral is also often observed from the surface of the $MoS_2$:Nb crystals, confirming their non-centrosymmetric stacking[26] (Supplementary Figure 3). Density functional theory (DFT) calculations were performed to confirm the stacking reconfiguration of $MoS_2$ from 2H phase in the updoped to 3R phase in the Nb-doped case. In Fig. 1b, the total energy difference between bilayer 2H and 3R $MoS_2$:Nb ($\Delta E_0$) turns into negative values, −0.13 to −0.41 meV per atom, depending on the Nb doping concentrations. We note that the 3R phase is energetically more stable than the 2H phase once the Nb dopant is added to $MoS_2$ supercell (the $6 \times 6$ ($4 \times 4$) supercell is employed for Ⓐ and Ⓑ (Ⓒ) doping configurations). And, it becomes increasingly more stable upon further Nb incorporation. In short, it is the thermodynamic driving force that reorders the stacking stability, i.e., the 3R phase will naturally form once the Nb dopants are included in the crystal growth at elevated temperatures of ~1000 °C for > 2 weeks. The relative energetic stability can be related to free-carrier screening by holes residing in the $d_{z^2}$ bands with some delocalization[23] in Nb-doped $MoS_2$, effectively lowering the total energy of non-centrosymmetric 3R-$MoS_2$. We note that the 3R phase was also experimentally observed in other heavily doped $MoS_2$[27]. Furthermore, the dynamic stability of such 3R $MoS_2$:Nb bilayer is confirmed by the phonon dispersion showing no imaginary part, a criteria employed for other poly types of $MoS_2$[28,29] (Supplementary Figure 4). Consequently, we conclude that Nb doping provides a reliable route of acquiring the uncommon, non-centrosymmetric structure of $MX_2$ bulk crystals for potential valleytronic applications, circumventing the need for large-area

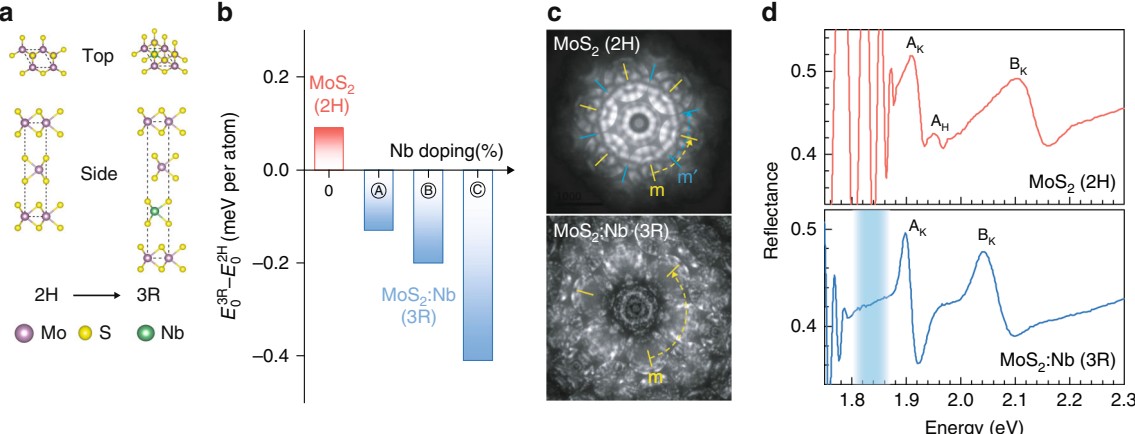

**Fig. 1** Crystal structure of undoped and Nb-doped bulk $MoS_2$. **a** Illustration of modification in stacking order polytypism of $MoS_2$ by Nb doping (top and side views). The dashed lines show the corresponding single unit cells of 2H and 3R structures. **b** Calculated energy difference of 3R phase with respect to 2H phase, with positive (negative) values indicating that the 2H (3R) phase is more stable. Bilayer $MoS_2$ system is considered for variable Nb doping concentrations (not to the scale). For the $MoS_2$:Nb, Ⓐ and Ⓑ are obtained with a $6 \times 6$ supercell by adding 1 and 2 Nb dopants, respectively. Type Ⓒ is calculated by substituting a Nb dopant into a $4 \times 4$ supercell. So, their Nb doping concentrations correspond to 1.4, 2.8 and 3.1%, respectively. **c** Convergent beam electron diffraction patterns obtained from the undoped and Nb-doped multilayer $MoS_2$ at [0001] incidence. Both bulk crystals display mirror symmetries, indicated by yellow and blue lines, for the six-fold ($MoS_2$, 2H) and three-fold ($MoS_2$:Nb, 3R), respectively. **d** Optical reflectance spectrum of bulk $MoS_2$ and $MoS_2$:Nb taken at 4.5 K. Two main peaks are assigned as A and B exciton transitions at the K point while the additional feature arising from H-point of the Brillouin zone is only observed in undoped $MoS_2$. The strong absorption below the A exciton in $MoS_2$:Nb is evidenced by the absence of the Fabry-Perot oscillation which appears in the Nb-free crystal

monolayers, nor delicate adjustment of the growth temperature gradient[25] (the routinely adopted means to stabilize 3R $MoS_2$).

The rearrangement of stacking order by the substitutional Nb doping also renormalizes the A and B exciton transitions at the K point of the Brillouin zone, as evidenced in optical reflectance spectra for $MoS_2$ and $MoS_2$:Nb bulk crystals measured at 4.5 K (Fig. 1d). First, the A exciton feature redshifts slightly by ~10 meV in the $MoS_2$:Nb crystal, determined by the lineshape fits to a Lorentz oscillator model[30] (Supplementary Figure 5 and Note 2). Second and more notably, the A–B energy splitting decreases from 0.19 to 0.15 eV, i.e., a much greater redshift of the B exciton peak. Also, the flatter dispersion along the K-H path in the 3R system[25] eliminates a distinctive H-point excitonic transition that is normally observable in bulk 2H-$MoS_2$ at low temperatures[30]. Quantitatively consistent with earlier reports on undoped 3R-$MoS_2$[25], it can be seen that these renormalization effects in the A and B exciton energies come from the 2H–3R structural transformation in the $MoS_2$:Nb, rather than a direct electronic effect associated with the Nb doping. Aside from these structural effects, strong and broad absorption is observed at 20–100 meV below the A exciton peak in the $MoS_2$:Nb crystal (Fig. 1d), and is attributed to Nb impurity states located above the valence band edge at the K point ($K_V$). Yet, as for what we confirm at the K point, the direct electronic influence of these Nb impurity states on the host $MoS_2$ bands is negligible, akin to the case of conventional bulk semiconductors in the heavy doping regime.

**Electronic restructuring of valence bands in the bilayer case.** Few-layer structures were obtained by mechanical isolation from the bulk crystals, which enables further visualization of effects of the different stacking sequences. Figure 2a displays high-resolution transmission electron microscopy (HRTEM) images of bilayers (2L) of undoped $MoS_2$ and $MoS_2$:Nb. Here the 2H stacking of undoped $MoS_2$ (top panel) is easily recognized with no atom located at the center of each hexagon. In the $MoS_2$:Nb bilayer, on the contrary, additional atomic columns appear in the center of each hexagon, with a slight but noticeable intensity

difference between nearest neighbor columns. Atomically precise positioning and stack ordering of layers are possible in such 3R-stacking bilayer with the aid of line intensity profile[31] (Supplementary Figure 6). With extensive HRTEM investigation, we confirm that structurally the Nb doping only induces the 3R polytypism in $MoS_2$, preserving its single crystallinity without causing extended defects and local lattice distortion.

Bulk $MoS_2$ is an indirect-bandgap semiconductor with negligible photoluminescence (PL), thus prevents access to its electronic bands in light emission spectroscopy. When the $MoS_2$ crystal is thinned to a few layers, however, noticeable PL emerges with two major features. A main peak at ~1.85 eV is associated with (hot-carrier) transitions across the direct bandgap at the K-valley, whereas a relatively weak PL peak at lower energies (Fig. 2b) arises from transitions across the indirect bandgap involving the $\Gamma_V$, which constitutes a combination of $Mo - d_{z^2}$ and $S - p_z$ orbitals. Since these orbitals extend along the z-direction with strong overlap between neighboring layers, the indirect PL peak acts as an indicator of interlayer coupling: the lower the indirect PL peak energy is, the stronger the interlayer coupling[32]. 3R stacking is known to have a slightly shorter interlayer spacing compared to that of 2H, so the indirect PL of 3R bilayers is expected to redshift. However, to separate the structural (2H vs. 3R) effect from electronic (doped vs. undoped) effect in Nb-doped 3R bilayers, it is necessary to also include Nb-free 3R bilayers in the comparison. Indeed, the indirect PL of Nb-free 3R bilayers redshifts slightly by ~30 meV, as shown in Fig. 2b. In contrast, the observed redshift of indirect PL from the Nb-doped 3R bilayers is much greater, up to ~140 meV, and it indeed monotonically redshifts further with Nb fraction ($x$). These effects suggest additional mechanism of Nb doping beyond the mere 2H–3R structural conversion.

The unusual evolution of indirect optical transition in 2L-$MoS_2$:Nb is attributed to a valence band modification by the Nb impurity states. As seen in Fig. 3, the impurity level of Nb replacing Mo (denoted as $E_I$) is theoretically known to be located below the $\Gamma_V$ in $MoS_2$ for both bulk and bilayer cases[33], also judged from the reflectance spectra in Fig. 1d, thus crossing and

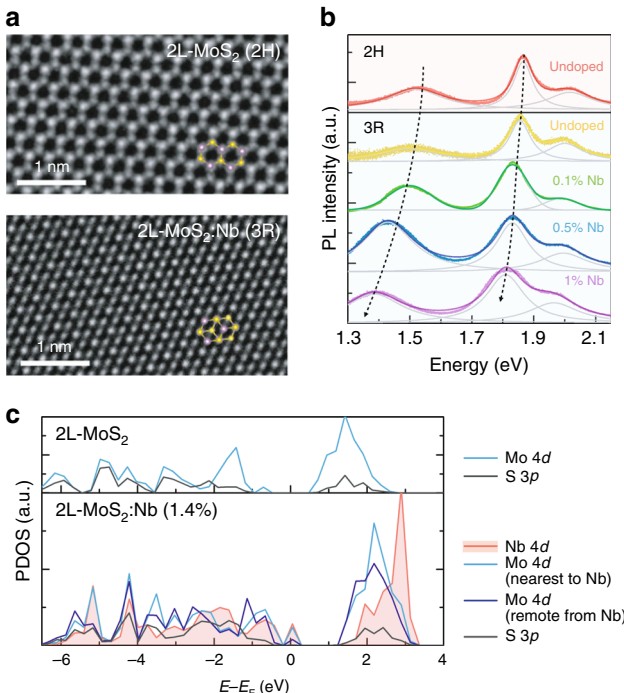

**Fig. 2** Beyond the 2H–3R structural transition: Restructuring of valence bands in bilayer Nb-doped MoS₂. **a** High-resolution transmission electron microscopy (HRTEM) images of typical bilayer (2L) MoS₂ and Nb-doped MoS₂ flakes. The insets show a modeled top view of 2H and 3R MoS₂ bilayers with violet and yellow spheres corresponding to Mo and S atoms, respectively, and are superimposed onto the HRTEM images. Here the existence of Nb dopants is not presented due to its substitutionality and indistinguishability from Mo in HRTEM ($Z_{Nb} = 41$ and $Z_{Mo} = 42$). Scale bars, 1 nm. **b** Room temperature photoluminescence (PL) spectra of 2L MoS₂ including both 2H and 3R structures, and MoS₂:Nb with three representative doping concentrations. The arrow line shows the evolution of indirect bandgap to guide the eye. **c** Calculated partial density of states (PDOS) of 2L undoped and Nb-doped MoS₂ (3R-stacked) projected onto the selected atoms, Mo, Nb and S. For the bilayer MoS₂:Nb, one of the 72 Mo atoms is substituted with a Nb dopant atom (corresponding to 1.4% doping) to match the experimental doping level. Also, two representative Mo atoms, the nearest to and remote away from Nb dopants, are shown here to resolve their distinct contributions to the valence band maximum

in resonance with the valence bands. In this case, an upshift of $\Gamma_V$ (hence a reduction of the indirect bandgap) is expected from a simple two-level repulsion model, as exemplified in similar cases of valence (conduction) bands of GaAs where As is partially substituted with Bi or Sb (N)[14,34]. More generally, this two-level band-anticrossing model has successfully described the electronic band restructuring in a wide range of substitutional semiconductor alloys[15,35,36]. It predicts the restructuring and hybridization of host bands ($E_V(\mathbf{k})$) with impurity states ($E_I$), giving rise to the two newly formed sub-bands expressed as

$$E_{\pm}(\mathbf{k}) = \begin{vmatrix} E_V(\mathbf{k}) & C\sqrt{x} \\ C\sqrt{x} & E_I \end{vmatrix} \quad (1)$$

where $C$ is a parameter describing the hybridization strength, and $x$ is the fraction of the impurities. For the case of bilayer MoS₂: Nb, $E_V(\mathbf{k})$ and $E_I$ are the original valence band dispersion of MoS₂ and the originally non-dispersive Nb energy level, respectively, and $E_{\pm}(\mathbf{k})$ is the two newly formed sub-bands as a result of the anticrossing interaction between $E_V(\mathbf{k})$ and $E_I$, as shown in Fig. 3a. Even a sub-1% doping causes a significant band

restructuring in MoS₂:Nb, suggesting a strong band-anticrossing interaction with a high interaction parameter, $C$.

We also performed ab initio calculations on the electronic structures using DFT within the local density approximation and the results confirmed this model. Figure 2c shows the partial density of states (PDOS) plot for 2L undoped and Nb-doped MoS₂ in the 3R-type stacking order. Upon Nb doping, the indirect bandgap becomes narrower and the Fermi level ($E_F$) downshifts consistent with our experimental observation. As seen from the undoped MoS₂, the band edges of both conduction and valence bands are mainly composed of Mo $4d$ states. For 2L-MoS₂:Nb, the Nb $4d$ state contributes most significantly to the valence band maximum, $\Gamma_V$, while its influence on the conduction band minimum is nearly negligible. Notably, contribution from the host Mo $4d$ state to the valence band edge is found to be sensitive to its distance to the Nb dopant; the closer to the Nb atom, the greater contribution of the Mo to the valence band maximum. This is not the case for the conduction band, hence it supports our band-anticrossing-type hybridization between the Nb and Mo $4d$ states at $\Gamma_V$, as outlined above. Atomic contribution-resolved band structures further elucidate the case by visualizing the hybridization at $\Gamma_V$ (Supplementary Figure 7). That is, as the doping concentration increases, the valence band maximum ($\Gamma_V$) gains more $E_I$ character from the Nb dopants simply because of the greater fraction of Nb, $x$; in the meantime, our calculation suggests that the interaction parameter, $C$, can be simultaneously enhanced due to the closer physical distance to the dopants as $x$ increases, all ultimately driving the indirect PL peak to monotonically redshift with $x$.

**Bound excitons in the monolayer limit.** When further thinning down to the monolayer limit, 2H- and 3R-MoS₂ no longer have a structural distinction (Supplementary Figure 8). From the bulk to monolayer, MoS₂ layers transit from indirect to direct bandgap, as well as experience reduced dielectric screening. These effects lead to well-known strong PL and large excitonic binding energy in monolayer MoS₂. For monolayer MoS₂, a drastic self-downshift of the $\Gamma_V$ point makes the $K_V$ pockets the valence band maximum, across which direct optical transitions take place. In such a system, the Nb impurity states are now located above the valence band maximums, no longer crossing the valence bands of MoS₂ (Fig. 3b) and hence becoming isolated in-gap impurity states. Such reconfiguration of relative location between the host valence bands and the impurity level results in a few notable changes.

First, we observed an enhanced, broad-band and redshifted PL emission at room temperature (RT) from the monolayer MoS₂: Nb, as shown in Fig. 4a. Combined with the corresponding absorption spectra (Supplementary Figure 9 and Note 3), it can be attributed to optically excited excitons (X) binding to neutral acceptor ($A^0$) resulting in the formation of $A^0X$ complexes. While an ionized acceptor ($A^-$) does not usually bind an exciton, the binding energy of exciton is in general the highest for a neutral acceptor according to the Haynes rule[37]. As illustrated in Fig. 3, the former ($A^-X$) and latter ($A^0X$) cases correspond to the bilayer and monolayer MoS₂:Nb, respectively, under the assumption that $E_I$ becomes a deep impurity level and a portion of acceptors are no longer thermally ionized for the monolayer MoS₂:Nb according to Fermi-Dirac statistics. In this sense, a direct PL of 2L-MoS₂:Nb is only weakly affected by the presence of $E_I$ (Fig. 2b) but that of monolayer MoS₂:Nb substantially changes. It is also noted that such impurity-bound PL becomes even broader and more redshifted upon increase in the Nb doping concentration. Next, PL of the monolayer MoS₂:Nb is seen to be more than an order of magnitude stronger than that of the undoped monolayer

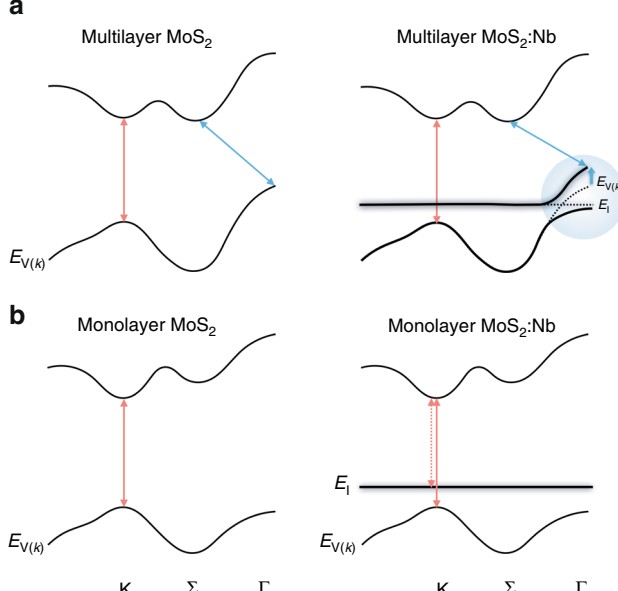

**Fig. 3** Band engineering of MoS₂ by Nb doping. Schematic band structures of **a** multilayer, such as bulk and bilayer, and **b** monolayer MoS₂ before (left panel) and after (right panel) degenerate Nb doping. For the case of multilayer MoS₂:Nb, the pristine valence bands of Nb-free MoS₂ are included as dotted black lines for a direct comparison, and the regions in the Brillouin zone ($\Gamma_V$) where a significant band restructuring takes place are also highlighted as a blue circle. Direct (impurity-bound) and indirect transitions are indicated by red (dotted) and blue arrow lines, respectively

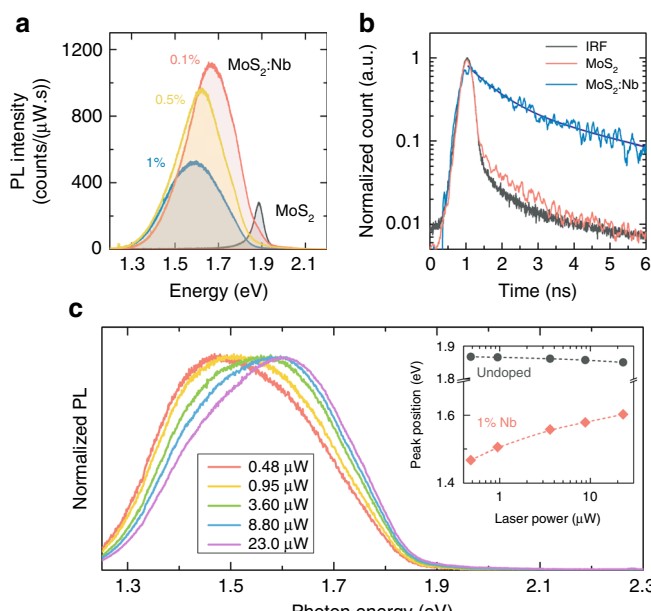

**Fig. 4** Photoluminescence in monolayer Nb-doped MoS₂. **a** Room temperature photoluminescence (PL) characteristics of monolayer MoS₂ and MoS₂:Nb. The spectra were collected under excitation with a 4.2 μW Ar-ion laser line (488 nm). **b** Time-resolved PL of monolayer MoS₂ and MoS₂:Nb. Black curve represents the instrumental response function (IRF). **c** A series of PL spectra with different excitation laser power for 1% Nb-doped MoS₂ monolayer. Inset shows the gradual PL peak shift of MoS₂:Nb upon increasing laser input power in comparison with undoped MoS₂ monolayer

MoS₂, with a much elongated PL lifetime as shown in Fig. 4b. That is, while the decay dynamics of the exciton states at ~1.88 eV in undoped MoS₂ is rather fast (lifetime, $\tau$, on the order of 100 ps[38]) approaching the instrumental response function (IRF), the PL at ~1.68 eV in the monolayer MoS₂:Nb shows significantly slower bi-exponential decay dynamics ($\tau_1 = 0.7 \pm 0.05$ ns and $\tau_2 = 4.3 \pm 0.3$ ns). This is in great contrast with the earlier PL studies on native defects like chalcogen atomic vacancies where PL decay times are measured as a few hundred ps[39,40]. Moreover, although such defect-related PL feature is observed only at cryogenic temperatures[24], here the Nb-induced sub-band PL is much brighter and stable even at RT. Doping could indeed, in some cases, enhance radiative efficiency and could, for example, even enable RT lasing in GaAs nanowires[41], yet such remarkable luminescence enhancement has not been reported in 2D semiconductors. Last, PL from undoped and Nb-doped monolayer MoS₂ shows a different laser excitation power dependence (Fig. 4c). As laser power steadily increases, PL in the monolayer MoS₂:Nb, stemming from the impurity-bound excitons as described above, monotonically shifts toward higher energy, as similarly observed for localized excitons[42] whilst the undoped MoS₂ monolayer only displays a slight decrease presumably due to laser heating effect.

Photoluminescence excitation (PLE) spectroscopy was also performed over a wide range of excitation energies to further understand this unusual PL behavior. It can be seen from Fig. 5 that the PLE signal appears when the excitation energy is above the A exciton absorption, and reaches a maximum when the excitation energy is equal to the B exciton absorption, as similarly reported in undoped monolayer MoS₂[43]. However, this enhancement of emission is not observed for excitation at C resonance indicating that an exciton first needs to form to bind to the impurities. Again, this PLE behavior is different from what is known for defect-related emission[44]. The fact that the PLE

emission starts only after the excitation energy is higher than the A exciton energy (rather than when the excitation energy is equal to the Nb impurity energy level), is also an indication that the Nb impurity states no longer form a dispersive band in this case (unlike at the $\Gamma_V$ band in multilayer or bulk). Therefore, the Nb states stay as non-dispersive energy levels not hybridizing with the $K_V$ host bands (Fig. 3b), instead contributing to the formation of bound excitons. Indeed, hybridization of impurity states with the host bands is expected to be much weaker at the Brillouin zone edge (K) than at the center ($\Gamma$). This is because the interaction strength ($C$) in the band anticrossing model becomes the maximum at $\mathbf{k} = 0$ (i.e., at $\Gamma_V$), with a rapidly decreasing $\mathbf{k}$-dependence that follows the Fourier transform of the spatially-decaying overlap integral between the impurity wavefunction and the Wannier function of the host bands[45]. Moreover, the hybridization at $K_V$ is further weakened due to the farther energy separation of $E_I$ from $E_V(\mathbf{k})$ at $K_V$ in the monolayer (than $\Gamma_V$ in multilayer/bulk).

## Discussion

We have demonstrated restructuring of layer stacking and electronic bands of MoS₂ by introducing substitutional Nb dopants. It drives the stacking sequence from ABABAB (2H) to ABCABC (3R), resulting in broken inversion symmetry even for multilayer MoS₂. Our study reveals that the Nb impurity states in the bulk are quite dispersive and strongly interact with the extended states of host valence band, which is also supported by theoretical analysis. Strong interaction mainly occurs at the $\Gamma_V$ of the Brillouin zone for multilayer or bulk MoS₂, suggesting a valence band hybridization. It is highlighted that all the structural and electronic modifications in multilayer MoS₂:Nb happen still within the relatively low substitutional fractions of doping (rather than

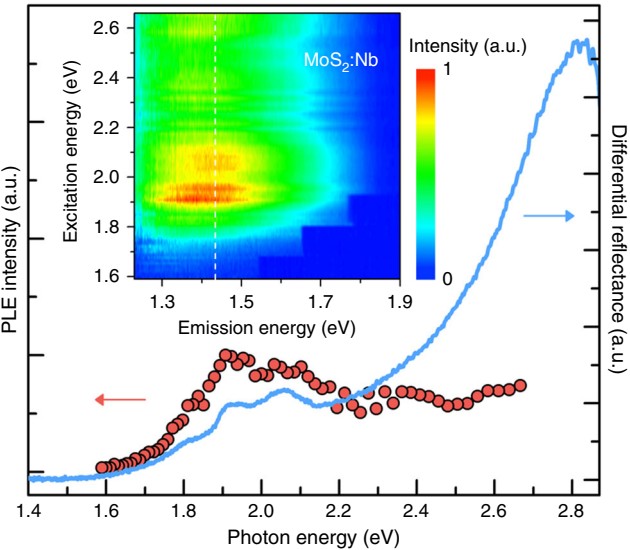

**Fig. 5** Photoluminescence excitation (PLE) data of the monolayer MoS₂:Nb (1%). Left and right y axis correspond to PL emission energy at 1.43 eV and differential reflectance spectra (ΔR/R), respectively, as a function of excitation energy. Top inset displays PLE intensity 2D map where the color scale represents emission intensity, and was collected with an excitation power of ~0.5 µW at 300 K

alloying). In contrast, in the case of monolayer, the Nb dopants acting as non-interacting impurity states significantly affect the excitonic transition along the K valley. Monolayer of MoS₂:Nb shows a distinctly strong, redshifted and long-lifetime PL monotonically depending on Nb doping concentrations. Our investigation is a step forward towards the possibility of enhancing optoelectronic performance and tuning excitonic effects in 2D semiconductors for potential applications in solid-state devices.

## Methods

**Materials preparation**. Undoped and Nb-doped MoS₂ single crystals were grown by a chemical vapor transport (CVT) method with the use of I₂ as a transport agent. For comparison, natural 2H-MoS₂ crystals were also purchased from SPI Supplies. High purity elements including Mo, S, Nb (all 99.99% purity), and the iodine transport agent were used for the crystal growth. The molar ratio of Mo/S was kept to 1:2 and the substitutional doping concentrations of niobium were designed to become 0.1, 0.5% and 1%, respectively. Afterwards, all reactants were placed in quartz tubes that were evacuated below ~2 × 10⁻⁵ Torr and sealed by an oxyhydrogen flame. A horizontal three-zone furnace was utilized to maintain an optimal temperature gradient for the diffusion of iodine transport agent. The quartz tubes were placed into the furnace where the high temperature zone was set at 1050 °C and the low temperature zone was set at 935 °C for 720 h.

**Structural characterization**. Both convergent beam electron diffraction (CBED) patterns and high-resolution transmission electron microscopy (HRTEM) images were acquired on a FEI Titan environmental TEM 80–300 at 80 kV accelerating voltage. A negative spherical aberration imaging technique with monochromated electron beam was used for HRTEM images of bilayer samples.

**Optical measurements**. The reflectance measurements on the bulk crystals used light from a 75 W Xenon lamp dispersed by a 1/2 m grating monochromator and a photo-multiplier tube detector, with the sample cooled using a continuous flow liquid He cryostat. Time-resolved photoluminescence measurements were performed using the time-correlated single-photon counting technique at room temperature. MoS₂ samples were excited with an ultrafast optical pulse of photon energy of 2.06 eV and power of 4.5 µJ/cm². Pulse duration and repetition rate were 50 ps and 20 MHz, respectively. The PL spectra were obtained with Raman spectrometers in the back scattering geometry and continuous wave 488 nm laser as the excitation source. For photoluminescence excitation (PLE) measurements, a Fianium supercontinuum white laser (SC450) coupled to a laser line tuneable filter (LLTF) was used to provide excitation sources. The excitation lasers, ranging from 460 nm to 760 nm, have a bandwidth of 1–2 nm and their power was kept below 0.5 µW for all the measurements in order to keep the excitation rate in the linear

regime and to avoid possible damage to the samples. Multiple measurements were made for each sample to check the reproducibility of the results.

**Electronic structure calculations**. Density functional theory (DFT) calculations were performed on the bilayer MoS₂ structures to elucidate their bandstructures before and after Nb doping. Geometry optimizations were performed using DFT implemented in the Vienna Ab initio Simulation Package (VASP)[46,47] within a Projected Augmented Wave (PAW)[48] basis and with the Perdew, Burke and Ernzerhof (PBE) functional[49], ensuring sufficient vacuum (at least 20 Å) between periodic images along the z-direction (perpendicular to the MoS₂ plane) to minimize spurious interactions. During structural optimization, all atomic coordinates and lattice vectors were fully relaxed until the absolute value of the forces acting on each atom was less than 0.01 eV/Å. We checked that sufficient vacuum remains after relaxation. Plane-wave cutoffs were set to 400 eV and van der Waals interactions were accounted for via the DFT-D2 scheme[50]. For both undoped and Nb-doped MoS₂, a 6 × 6 supercell was used along with a 4 × 4 × 1 **k**-point grid, unless otherwise stated, with the Monkhorst-Pack sampling[51].

**Data availability**. The data that support the findings of this study are available from the corresponding authors on request.

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

## Acknowledgements

This work was supported by the Director, Office of Science, Office of Basic Energy Sciences, Materials Sciences and Engineering Division, of the U.S. Department of Energy under Contract No. DE-AC02-05CH11231. W.W. and Y.C. acknowledge support from the Singapore-Berkeley Research Initiative for Sustainable Energy (SinBeRISE). G.E. acknowledges Singapore National Research Foundation, Prime Minister's Office, Singapore, for funding the research under its Medium-sized Centre program as well as NRF Research Fellowship (NRF-NRFF2011-02). T.L.T. acknowledges the use of high-performance computing facilities in A*STAR Computational Resource Centre (ACRC) in Singapore for the DFT computations performed in this work. D.-Y.L acknowledges the financial support from the Ministry of Science and Technology of Taiwan, Republic of China under contract No. MOST 105-2112-M-018-006. We thank Prof. Sefaattin Tongay and Dr. Can Ataca for helpful discussions in an early stage of the work. Prof. Y. Iwasa and R. Suzuki are also highly appreciated for providing undoped 3R $MoS_2$ crystals.

## Author contributions

J.S and J.W. conceived and supervised the project. J.S., W.Z., T.-E.P. and G.E. fabricated the samples, conducted the optical and electrical measurements, and analyzed the data. J. K. C.J. and F.W carried out TRPL measurements. N.S. and S.G. helped the reflectance measurements. T.L.T. and Z.M.W. performed the theoretical calculations. D.-Y.L. grew the samples. J.P. performed TEM characterizations. J.S. and J.W. wrote the manuscript. All authors discussed the results and commented on the paper.
