## [Peer Review File · Nature Communications]

Reviewers' comments:

Reviewer #1 (Remarks to the Author):

The work has shown the structure transformation after with Nb doping. It seems a following work of reference 6. However, from reference 6, it did not show 3R structural crossover. Hence, the EXAFS may still be needed to identify whether Nb has been successfully replace Mo site. In addition, Nb doped MoS₂ have been commercially available, but it does not show 3R structure. Hence, the authors have to provide the mechanism behind the phenomenon. It is a requirement for a high impact journal like Nature Communications.

For the first principle calculation, the author used PBE functional. It belongs to GGA method. It is well known that first principle calculations have many functionals. For the publication in Nature Communication, GGA is not enough, GGA+U, hybrid function are required to confirm the GGA calculation. As I said previously, Nb doped MoS₂ commercially available, but without 3R transformation. The accuracy of calculation is more important. In addition, the calculation doping concentration is 3%, larger than 1% doping in experiment. Larger supercell is needed to do the calculation, consistent with experiment conditions. The cutoff energy is also relatively low.

Reviewer #2 (Remarks to the Author):

The study reported in this article, describes both Nb-free and Nb-doped MoS₂ single crystals (synthesized using the chemical vapour transport methods). The authors further perform DFT calculations to confirm the observed doping-induced stacking reconfiguration of MoS₂. Where as the results are promising, the following concerns needs to be addressed before I can recommend the manuscript for publication in Nature Communications.

In this study authors manifested the effects of substitutional doping (with "Nb") on structural conversion from 2H to 3R. They have further mentioned

a) "Density functional theory (DFT) calculations were performed to confirm the observed doping-induced stacking reconfiguration of MoS₂. To be specific, ΔE_0 turns into -0.41 meV per atom between bilayer 2H and 3R MoS₂: Nb, implying that the 3R phase becomes energetically more favourable when a single Nb dopant is added to the 4×4 supercell (see Methods and Supplementary Table 1)."

b) "In contrast, the observed redshift of indirect PL from the Nb-doped 3R bilayers is much greater, up to ~ 140 meV, and it indeed monotonically redshifts further with Nb fraction (x). These effects suggest additional mechanism of Nb doping beyond the mere 2H-3R structural conversion."

1)

However looking into Fig. 1 (a), it is not at all clear, how the 2H to 3R conversion is taking place. Authors should illustrate all the intermediate stages (2H \rightarrow 3R). Nonetheless they must show, whether the final geometry optimized structure exactly replicates 3R or, not. In order to prove the stability of the final structure, they should also emphasize on the dynamical stability part (comparing both the Nb-doped and the Nb-free cases). Such studies, for other polytypes of MoS₂ are already available in literature. See the following Refs. :

[1] Science 12 Dec 2014: Vol. 346, Issue 6215, pp. 1344-1347 DOI: 10.1126/science.1256815

[2] Appl. Phys. Lett. 108, 253106, 2016, doi: <http://dx.doi.org/10.1063/1.4954257>

2)

The bandstructure diagrams of Fig. 3 should include more details. Specially, as the authors claim different phenomena at the V.B. max and C.B. min (i.e., in one case the "Mo" 4d states are dominant, whereas for the other one there is valence-band hybridization), the contribution resolved projected band structures could be immensely helpful for better understanding. Nevertheless, Authors mentioned

"As seen in Figure 3, the impurity level of Nb replacing Mo (denoted as EI) is theoretically known to be located below the Γ V in MoS₂ for both bulk and bilayer cases²⁷, also judged from the reflectance spectra in Figure 1c, thus crossing and in resonance with the valence bands."

3)

Did the study of Ref. 27 discuss anything regarding the "Nb" doping of bi-layer MoS₂ (or, even few layers)? Authors emphasized on the nonlinear modification the electronic band structure (while doped with "Nb"), but it would be better if they also cite a few theoretical studies showing the effects of both n-type-doping and p-type-doping (in the context of MoS₂).

Another concern with this article is the methodology part that describes electronic structure calculations.

They have said

"During relaxation, the supercell volume remains fixed, although its shape is allowed to change. We checked for sufficient vacuum after relaxation. Plane-wave cutoffs were set to 400 eV and all atomic coordinates and lattices were fully relaxed until the absolute value of the forces acting on each atom was less than 0.01 eV/Å. Spin-orbit coupling was turned on and van der Waals interactions were accounted for via the DFT-D2 scheme⁴⁴. For the undoped MoS₂, a 1 × 1 unit cell was used along with a 27 × 27 × 1 k-point grid with Monkhorst-Pack sampling⁴⁵. For the Nb-doped MoS₂, a 4 × 4 supercell was used with a 7 × 7 × 1 k-point grid. Two doping compositions were explored for the bilayer MoS₂ with 3R symmetry, by substituting up to 2 Mo atoms (with Nb) in the 4 × 4 supercell. For computation of electronic density of states (DOS), calculations were performed with the atomic positions fixed at the optimized structures and with a denser k-point mesh (15 × 15 × 1 k-point grid for the 4 × 4 supercell). Spin-orbit coupling was turned off for the DOS calculations.

4) What is the purpose of turning on SOC here? Did they show spin-resolved calculation?

5) While comparing two systems, one should not vary the super cell sizes and k-point grids randomly. Are these values, "1x1 cell with 27x27x1 k-points for undoped MoS₂ and 4x4 super cell with 7x7x1 k-points" adopted for geometry optimization? Better, the authors only mention the super cell sizes and the k-point grids which have been used for electronic structure calculations. All the super cells should be of same size (say 4x4), for any fair comparison (So the k-point grids).

6) What does this mean "supercell volume remains fixed, although its shape is allowed to change"? Does the geometry optimization consider stress-optimization as well (or, the lattice constants are fixed)? Otherwise, the structure might be at a saddle point.

7) The PDOS for bi-layer MoS₂ is shown in Figure 2 (c). For such intrinsic sample, why the energy zero (E-EF) line is shifted more towards the valance band edge? Authors must verify the value of k-point grid used (though they have not derived any quantitative information from these PDOS diagrams).

General Responses to Referees' Comments

We thank both referees for carefully reviewing our manuscript and providing a series of constructive comments. In general, both referees agree on the novelty and importance of our work. However, they also expressed concerns for the computational methodology of DFT calculations that support our experimental finding (Reviewer 1, 2) as well as the structural transition (Reviewer 1), so asked relevant questions with details and provided some suggestions. We considered these comments very seriously, and thereby we have obtained **seven additional sets of new experimental and theoretical data and subsequently performed new analysis**, all of which support the central claim of our manuscript: *reconstruction of stacking order and electronic bands of MoS₂ by substitutional doping*. Below, we present the point-by-point responses to the referees' comments, followed by summary of relevant changes (at the end of sub-section) and a reference list cited in the revision (at the end of this letter). We have thoroughly revised our manuscript based on our new experimental data and theoretical understanding as described in the summary of changes. These additions have substantially improved our manuscript, so we hope that you find it ready for publication at *Nature Communications*. We are deeply grateful for both referees for contributing to strengthening the manuscript with their insightful comments.

Sincerely,

Junqiao Wu & Joonki Suh

Point-by-Point Responses to Reviewer #1's Comments

R1-1. *The work has shown the structure transformation after with Nb doping. It seems a following work of reference 6. However, from reference 6, it did not show 3R structural crossover. Hence, the EXAFS may still be needed to identify whether Nb has been successfully replace Mo site. In addition, Nb doped MoS₂ have been commercially available, but it does not show 3R structure. Hence, the authors have to provide the mechanism behind the phenomenon. It is a requirement for a high impact journal like Nature Communications.*

We thank the reviewer for giving us an opportunity to clarify a few crucial points. Our current manuscript is a much advanced study following our own earlier work, [Ref #6: Suh *et al.* Nano Lett. 14, 6976 (2014)], as the reviewer pointed out. While the previous work merely showcases heavily *p*-type substitutional doping of MoS₂ at a single doping concentration, our current study systematically presents how Nb dopants alter the crystal and electronic structures distinctively for different thicknesses and at various doping concentrations. Structural data such as high-resolution transmission electron microscopy (HRTEM) images in Ref #6 was acquired from a monolayer flake for which the 2H and 3R stacking no longer make a difference. This was already clearly stated in our main text (Page 6, 3rd paragraph), and yet we conducted additional HRTEM characterization to directly support our argument. In

Figure R1. High-resolution transmission electron microscopy (HRTEM) images with sub-Å resolution acquired from Nb-doped and Nb-free monolayers, respectively. Both scale bars correspond to 1 nm.

addition to HRTEM images taken from the monolayer samples as displayed in Figure R1, none of characterization techniques including XPS, EF-TEM, and charge transport measurements employed in Ref #6 were capable of discerning how Nb dopants alter the crystal and electronic structures of MoS₂ crystal, a key finding of our current study. Hence, there is no contradiction between this study and earlier study.

Performing extended X-ray absorption fine structure (EXAFS) measurements is a good suggestion but not practical since such synchrotron-based characterization needs a pre-approved user proposal. In fact, it is also unnecessary as we have previously proved the substitutional Nb doing in MoS₂, and different doping levels are not expected to affect its substitutional nature [Ref. R1: F. A. Argül, Composites Part B 91, 589 (2016)]. However, we still performed new Hall effect measurements

Figure R2. (a) Room-temperature hole concentration and mobility as a function of Nb doping density. Hall-effect data for 0.5% Nb doping were imported from Ref #6. (b) Temperature dependent carrier concentration and mobility for 1% Nb-doped MoS₂. Inset scale bar, 10 μm.

density is in quantitative agreement with the nominal doping level. For example, 1 atomic % Nb doping introduces holes at a density of $\sim 1.3 \times 10^{20} \text{ cm}^{-3}$, which matches the estimated value using the Mo atomic volume of $1.9 \times 10^{22} \text{ cm}^{-3}$ in 3R-MoS₂ [based on lattice parameters in Ref. R2: S. Anghel et al. arXiv:1411.3850 (2014)], assuming single acceptor character of Nb when replacing Mo [Ref #18]. Therefore, our new Hall-effect data simultaneously prove the substitutionality and successful incorporation of Nb dopants in MoS₂ for the three doping levels studied.

As for the commercial product, our team has searched and contacted most of the major crystal suppliers including 2D Semiconductors, HQ Graphene, SPI, and MTI Corporation. It was found that only HQ Graphene has Nb-doped MoS₂ crystal in stock, but at one fixed doping concentration without specifying the structural phase. Moreover, simple techniques of characterization, such as SEM, EDS and Raman spectra normally provided by the vendor, are not reliable means to conclude the stacking order of bulk MoS₂ crystals owing to the too subtle difference between the 2H and 3R structures. That is why we (and Ref. #21) have relied on convergent beam electron diffraction (CBED) and absorption spectra to identify the structural phase.

Figure R3. Statistics of the crystal structure of undoped and Nb-doped MoS₂ single crystals determined by convergent beam electron diffraction from a total of 51 samples.

Nevertheless, taking the reviewer's comment very seriously, we have carried out more CBED measurements for a more compelling conclusion. A total of 51 crystal samples have been imaged and analyzed; while all of natural undoped MoS₂ flakes are determined as the 2H phase, it is clear that the degenerately Nb-doped MoS₂ mostly takes the 3R phase (Figure R3). Only a small fraction of the Nb-doped MoS₂ flakes (3 out of 42, ~7%) exhibits the 2H stacking, and this small fluctuation does not invalidate our conclusion of the stacking restructuring, given the fact that even undoped MoS₂ crystals were reported to consist of ~10% 3R or mixed phase of 2H and 3R [Ref. R3: J.-U. Lee et al. ACS Nano 10, 1948 (2016)].

Our new DFT calculations reveal that the relative change in the ground-state total energy of the two phases causes the 2H-to-3R structural conversion in Nb-doped MoS₂ (please see Figure R5 in the following section R2-1, and note that the DFT calculation were re-done with a 6 × 6 supercell). Given the fact that the 3R phase was also observed in MoS₂ containing some other impurities [Ref. R4: K. K. Tiong et al. J. Crystal Growth 205, 543 (1999) and Ref. R5: R. J. Traill, Can. Mineral. 7, 524 (1963)], adding considerable amount of dopants tend to stabilize the low-symmetry rhombohedral structure; but indeed, more follow-up studies are required to fully understand this effect in a systematic way. Our work serves to catalyze such experimental and theoretical studies as doping and alloying in 2D semiconductors become an active research area.

Relevant changes made:

- Additional HRTEM figure (Supplementary Figure 8) is now presented to confirm no structural distinction between undoped and Nb-doped MoS₂ in the monolayer limit.
- New Hall effect data are added as Supplementary Figure 1, and the relevant discussion is added in the main text (Page 4, 2nd paragraph).
- A few sentences are added to provide statistical information about our extended CBED characterization (Page 4, 2nd paragraph in the main text and Supplementary Note 1).
- We include a new statement about impurity-related structural conversion in MoS₂. Two new references are now added (Page 4, 2nd paragraph).

R1-2. *For the first principle calculation, the author used PBE functional. It belongs to GGA method. It is well known that first principle calculations have many functionals. For the publication in Nature Communication, GGA is not enough, GGA+U, hybrid function are required to confirm the GGA calculation. As I said previously, Nb doped MoS₂ commercially available, but without 3R transformation. The accuracy of calculation is more important. In addition, the calculation doping concentration is 3%, larger than 1% doping in experiment. Larger supercell is needed to do the calculation, consistent with experiment conditions. The cutoff energy is also relatively low.*

We appreciate the suggestions on our DFT part. Most importantly, we have re-done all calculations mostly with a larger, 6 × 6 supercell, expanded from the previously utilized 4 × 4 supercell, unless otherwise stated. It enables us to achieve as low as 1.4% Nb doping by substituting 1 Nb dopant with one of the 72 Mo host atoms in bilayer MoS₂, substantially reducing the gap between experimental and theoretical doping levels. Under this new computational condition, we confirmed that our previous conclusion from DFT calculations still remains valid for multilayer Nb-doped MoS₂ as shown in Figure R4: *i*) electronically, the Nb 4d state contributes most significantly to the valence band maximum (VBM) while contribution from the host Mo 4d state to the valence band edge is found to be sensitive to its distance to the Nb dopant, supporting the exclusive orbital

Figure R4. Calculated partial density of states (PDOS) of 2L undoped and Nb-doped MoS₂ (3R-stacked) projected on the selected atoms, Mo, Nb and S. For the bilayer MoS₂:Nb, two representative Mo atoms, the nearest to and remote away from Nb dopants, are shown here considering their distinct contribution to the valence band maximum.

overestimates the monolayer MoS₂ bandgaps by up to 0.4 eV [Ref. R6: Y. Jing et al. J. Mater. Chem. A 2, 16892 (2014) and Ref. R7: D. Liu et al. Appl. Phys. Lett. 103, 183113 (2013)], and U parameter value for MoS₂ is currently under-developed for GGA+U method. Furthermore, a recent DFT study shows that the relative orbital contributions to band structure of MoS₂ are independent of the functional used [Ref. R8: J. Su et al. RSC Adv. 5, 68085 (2015)]. These analyses justify our use of the PBE/GGA methods, instead of the other very expensive computational approaches, as similarly argued in other *Nature Communications* work [Ref. R9: A. P. Nayak et al. Nat. Commun. 5, 3731 (2014)]. Lastly, the energy cutoff value of 400 eV is in fact not small, and we have thoroughly tested it previously with a higher cutoff of 500 eV, and we found that it converges to within 0.002 eV.

Relevant changes made:

- Figure 2(c) and the relevant discussion in the main text (Page 6, 2nd paragraph) are updated based on new DFT calculations with a larger supercell. Corresponding figure caption is also rewritten indicating that the new DFT condition is very close to experimental doping density.
- Method section describing computational details is also modified accordingly.

Point-by-Point Responses to Reviewer #2's Comments

The study reported in this article, describes both Nb-free and Nb-doped MoS₂ single crystals (synthesized using the chemical vapour transport methods). The authors further perform DFT calculations to confirm the observed doping-induced stacking reconfiguration of MoS₂. Whereas the results are promising, the following concerns needs to be addressed before I can recommend the manuscript for publication in Nature Communications.

We thank the referee for supporting the manuscript and providing us with very useful comments and suggestions. In the following we address each point raised by the referee.

In this study authors manifested the effects of substitutional doping (with “Nb”) on structural conversion from 2H to 3R. They have further mentioned a) “Density functional theory (DFT) calculations were performed to confirm the observed doping-induced stacking reconfiguration of

hybridization between the Nb and Mo 4d states at the VBM, and *ii*) structurally, the 3R phase is energetically favourable thanks to lower ground-state total energy (see more details in the following R2-1 section).

It is also suggested to try a new methodology for the first principle calculations such as GGA+U and/or hybrid functionals. Such higher level DFT, however, does not always guarantee more accurate prediction, especially for 2D layered materials. Representatively, it has been known that hybrid functional (HSE)

MoS₂. To be specific, ΔE_0 turns into -0.41 meV per atom between bilayer 2H and 3R MoS₂:Nb, implying that the 3R phase becomes energetically more favourable when a single Nb dopant is added to the 4×4 supercell (see Methods and Supplementary Table 1).” b) “In contrast, the observed redshift of indirect PL from the Nb-doped 3R bilayers is much greater, up to ~ 140 meV, and it indeed monotonically redshifts further with Nb fraction (x). These effects suggest additional mechanism of Nb doping beyond the mere 2H-3R structural conversion.”

R2-1. However looking into Fig. 1 (a), it is not at all clear, how the 2H to 3R conversion is taking place. Authors should illustrate all the intermediate stages (2H \rightarrow 3R). Nonetheless they must show, whether the final geometry optimized structure exactly replicates 3R or, not. In order to prove the stability of the final structure, they should also emphasize on the dynamical stability part (comparing both the Nb-doped and the Nb-free cases). Such studies, for other polytypes of MoS₂ are already available in literature. See the following Refs.: [1] Science 12 Dec 2014: Vol. 346, Issue 6215, pp. 1344-1347 DOI: 10.1126/science.1256815 and [2] Appl. Phys. Lett. 108, 253106, 2016, doi: <http://dx.doi.org/10.1063/1.4954257>.

We thank the referee for the suggestion to include further discussion of the 2H-3R structural conversion our manuscript. First of all, we experimentally confirmed that the final geometry that Nb-doped MoS₂ has taken is exactly 3R stacking by a combination of convergent beam electron diffraction (main Fig. 1c and supplementary Fig. 2) and high-resolution transmission electron microscopy (main Fig. 2a and supplementary Fig. 6). It has been reportedly known that a strong interlayer Coulomb repulsion may destabilize other possible stacking orders than the 2H and 3R.

Our conclusion is further supported by new DFT calculations performed with larger and various supercells as presented in Figure R5, where relative difference in ground-state total energies of 2H and 3R phases is plotted as a function of Nb doping density in which three different Nb doping configurations are considered and arranged by the doping amount (but not scaled). We found that the 3R phase is energetically more stable than the 2H phase once Nb dopant is

Figure R5. Calculated total energy difference between the 2H and 3R phases of bilayer MoS₂ for variable Nb doping density. For the MoS₂:Nb, (A) and (B) are obtained with a 6×6 supercell by adding 1 and 2 Nb dopants, respectively. Type (C) is calculated by inserting a Nb dopant into a 4×4 supercell. Their Nb doping concentrations correspond to 1.4%, 2.8% and 3.1% respectively.

added to MoS₂, and it becomes increasingly more stable upon further Nb incorporation. In this context, we also note that 3R is more frequently obtained for layered NbS₂ crystals [Ref. R10: R. M. A. Lieth and J. C. J. M. Terhell, Transition Metal Dichalcogenides in *Preparation and Crystal Growth of Materials with Layered Structures* edited by R. M. A. Lieth, Springer Science, 1977].

Secondly, we prove the stability of such 3R structure of MoS₂:Nb emphasizing on the dynamical stability taking the referee’s comment seriously. The phonon calculations of the relaxed Nb-doped MoS₂ 4×4 supercell were carried out using the PHONOPY code [Ref. R11: A. Togo and I. Tanaka, Scr. Mater. 108, 1 (2015)], using the finite atomic displacements method with an amplitude of 0.01 Å to obtain the atomic forces within the supercell. This is followed by the dynamical matrix approach to acquire the phonon frequencies. In Figure R6, we confirmed that the phonon dispersion of 3R-MoS₂:Nb bilayer displays no imaginary part, supporting its dynamic stability as discussed in the references that the referee mentioned.

Figure R6. Phonon dispersion and corresponding phonon density of states (DOS) of the Nb-doped 3R-MoS₂ supercell.

potential barrier for rotating the 3R into the 2H structure [Ref. R12: A. Shmeliov et al. ACS Nano 8, 3690 (2014)].

Relevant changes made:

- Figure 1(b) in the main text is newly inserted and used when discussing the structural transformation happened during the Nb-doped MoS₂ synthesis. Corresponding figure caption and descriptive text (Page 4, 2nd Paragraph) are added too.
- Phonon dispersion is added as Supplementary Figure 4 for the dynamic stability along with a new sentence in the main text (Page 4, 2nd Paragraph).

R2-2. *The bandstructure diagrams of Fig. 3 should include more details. Specially, as the authors claim different phenomena at the V.B. max and C.B. min (i.e., in one case the “Mo” 4d states are dominant, whereas for the other one there is valence-band hybridization), the contribution resolved projected band structures could be immensely helpful for better understanding. Nevertheless, Authors mentioned “As seen in Figure 3, the impurity level of Nb replacing Mo (denoted as E_I) is theoretically known to be located below the Γ_V in MoS₂ for both bulk and bilayer cases, also judged from the reflectance spectra in Figure 1c, thus crossing and in resonance with the valence bands.”*

We sincerely thank the referee for this suggestion, and we carried out new electronic structure calculations. Figure R7 shows the contribution resolved band structures of 3R bilayer Nb-doped MoS₂ with projection onto the Nb dopant and Mo host atomic states. While the degree of influence from each atom on the conduction band minimum (CBM) appears quite similar, atomic contributions to the valence band maximum (VBM), Γ_V , are highly dependent on the constituent atoms. Most notably, Nb dopant is the one most responsible for the VBM, and the contribution of the host Mo to the VBM is found to be sensitive to its distance to the Nb dopant; the closer to the Nb atom, the greater contribution of the Mo to the VBM. This is not the case for the CBM, hence it supports our band-anticrossing-type hybridization between the Nb and Mo atomic states at Γ_V .

Our crystal growth of MoS₂:Nb takes place under a thermodynamically equilibrium condition (at elevated temperature for >1 week), so it may simply take the most energetically stable structure during the synthesis (Figure R5). In this sense, our schematic drawing in Figure 1a clearly illustrates an important role of Nb dopant as a main driving source toward 3R-MoS₂ (we confirmed that usual 2H-MoS₂ were obtained under the identical growth condition in the absence of the Nb source). Thus, it is quite challenging to define the intermediate states between the 2H and 3R, unlike other structural phase transition triggered by external stimuli. We also believe that any possible post-growth structural transformation, *i.e.*, back to the 2H phase, is prohibited by the substantial

Our current schematics in Figure 3 certainly help the readers understanding this phenomenon, so we decided to keep this while adding Figure R7 as a separate figure in supplementary information.

Figure R7. Band structures of bilayer 3R Nb-doped MoS₂ projected to Nb and Mo atoms. For Mo, the projections are made on two atoms with different distances from Nb dopant for comparison. A 4 × 4 supercell is used, and it is found that $E_F = -0.35$ eV. The scale indicated the magnitude of the projection.

Relevant changes made:

- Figure S7, projected band structures, is created, and a supporting sentence is added in the main text (Page 6, 2nd paragraph).

R2-3. Did the study of Ref. 27 discuss anything regarding the “Nb” doping of bi-layer MoS₂ (or, even few layers)? Authors emphasized on the nonlinear modification the electronic band structure (while doped with “Nb”), but it would be better if they also cite a few theoretical studies showing the effects of both n-type-doping and p-type-doping (in the context of MoS₂).

No, Ref #27 only investigated monolayer Nb-doped MoS₂ concentrating on the electrical contact behaviour with Au electrodes. Still, it supports our experimental observations and understanding in terms of *i*) location of the acceptor state of Nb dopants and *ii*) minor variation in the direct-bandgap of monolayer Nb-doped MoS₂.

Until now, the majority of theoretical doping studies for MoS₂ have only focused on its feasibility and introduction of free charge carriers, listing Nb (Re) as the most promising substitutional *p(n)*-type dopant [Ref #18, Ref #27, and Ref. R13: K. Dolui et al. Phys. Rev. B 88, 075420 (2013)]. Only a few recent *ab initio* studies investigate the other effects of doping MoS₂, e.g., ferromagnetic property of Re-doped MoS₂ [Ref. R14: P. Zhao et al. Comput. Mater. Sci. 128, 287 (2017)]. So we hope our work inspires future studies into broader topics of doping 2D materials both in theoretical and experimental approaches.

Relevant changes made:

- A new sentence is added to the main text (Page 3, 1st paragraph), and three new theoretical papers are cited accordingly.

Another concern with this article is the methodology part that describes electronic structure calculations. They have said “During relaxation, the supercell volume remains fixed, although its shape is allowed to change. We checked for sufficient vacuum after relaxation. Plane-wave cutoffs were set to 400 eV and all atomic coordinates and lattices were fully relaxed until the absolute value of the forces acting on each atom was less than 0.01 eV/Å. Spin-orbit coupling was turned

on and van der Waals interactions were accounted for via the DFT-D2 scheme. For the undoped MoS₂, a 1 × 1 unit cell was used along with a 27 × 27 × 1 k-point grid with Monkhorst-Pack sampling. For the Nb-doped MoS₂, a 4 × 4 supercell was used with a 7 × 7 × 1 k-point grid. Two doping compositions were explored for the bilayer MoS₂ with 3R symmetry, by substituting up to 2 Mo atoms (with Nb) in the 4 × 4 supercell. For computation of electronic density of states (DOS), calculations were performed with the atomic positions fixed at the optimized structures and with a denser k-point mesh (15 × 15 × 1 k-point grid for the 4 × 4 supercell). Spin-orbit coupling was turned off for the DOS calculations.

R2-4. *What is the purpose of turning on SOC here? Did they show spin-resolved calculation?*

We greatly appreciate the series of questions/suggestions to clarify our computational methodology. During our initial computation, we turned on spin-orbit coupling (SOC) to detect whether the relativistic effects introduce a splitting of the electronic bands, and indeed found it to be small. Moreover, SOC calculations are computationally more expensive and infeasible for large unit cells simultaneously considering Reviewer #1's suggestion for a larger supercell. Thus, we have now decided to show only non-SOC results after a revision process, and corrected the description accordingly.

Relevant changes made:

- Electronic structure calculations part in the Methods section is modified to clarify that no SOC results are presented.

R2-5. *While comparing two systems, one should not vary the super cell sizes and k-point grids randomly. Are these values, “1x1 cell with 27x27x1 k-points for undoped MoS₂ and 4x4 super cell with 7x7x1 k-points” adopted for geometry optimization? Better, the authors only mention the super cell sizes and the k-point grids which have been used for electronic structure calculations. All the super cells should be of same size (say 4x4), for any fair comparison (So the k-point grids).*

We totally agree on this point. Upon Reviewer #1's suggestion, we have recalculated for both cases, Nb-doped and Nb-free MoS₂ under the identical computational conditions using a 6 × 6 supercell with the set of k-points (4 × 4 × 1) for a fair comparison.

Relevant changes made:

- Methods part is updated to indicate our new computation conditions of supercell and k-point sizes.

R2-6. *What does this mean “supercell volume remains fixed, although its shape is allowed to change”? Does the geometry optimization consider stress-optimization as well (or, the lattice constants are fixed)? Otherwise, the structure might be at a saddle point.*

We confirmed that the structures are indeed relaxed fully, *i.e.*, both atomic positions and lattice constants are relaxed, to achieve a fully optimised stable structure. When simulating 2D materials such as MoS₂ in VASP, it is customary to introduce a large amount of vacuum in the supercell along the z-direction. To prevent a “collapse” of the vacuum regions, potentially arising from attractive forces between periodic images of the MoS₂, one can avoid optimising all 3 lattice vectors which is normally done for traditional bulk materials. For 2D materials, we can afford to

constrain one degree of freedom, and the degree of relaxation is indeed small and the amount of vacuum do not change much as long as the starting unit cell and atomic positions are reasonably set [Ref. R15: T. L. Tan et al. *J. Phys. Chem. C* **120**, 2501 (2016)]. We have always checked those factors after optimization, and confirmed that the stresses are very small and sufficient vacuum still exists. If the vacuum region is insufficient, one can always restart a calculation with a larger vacuum region in the supercell.

Relevant changes made:

- We rewrite the corresponding sentence in Methods section taking referee's suggestion: During structural optimization, starting from a large fixed-volume supercell with sufficient vacuum (in the z -direction) to prevent spurious interactions between periodic images, all atomic coordinates and lattice vectors were fully relaxed until the absolute value of the forces acting on each atom was less than 0.01 eV/Å. We checked that sufficient vacuum remains after relaxation.

R2-7. *The PDOS for bi-layer MoS₂ is shown in Figure 2 (c). For such intrinsic sample, why the energy zero ($E-E_F$) line is shifted more towards the valance band edge? Authors must verify the value of k -point grid used (though they have not derived any quantitative information from these PDOS diagrams).*

We have re-plotted the PDOS of bilayer MoS₂ using the newly calculated results from a 6×6 supercell with the set of k -points ($4 \times 4 \times 1$). We can confirm that Fermi level is located closer to the conduction band minimum implying a moderate n -type semiconductor when MoS₂ is not extrinsically doped.

Relevant changes made:

- Figure 2c is updated with new DFT results and a new sentence is added (Page 6, 2nd paragraph).

Revision References

- R1. F. A. Akgül, Influence of Ti doping on ZnO nanocomposites: synthesis and structural characterization. *Composites Part B* **91**, 589–594 (2016).
- R2. S. Anghel *et al.* Identification of 2H and 3R polytypes of MoS₂ layered crystals using photoluminescence spectroscopy. arXiv:1411.3850 (2014).
- R3. J.-U. Lee *et al.* Raman signatures of polytypism in molybdenum disulfide. *ACS Nano* **10**, 1948–1953 (2016).
- R4. K. K. Tiong *et al.* Growth and characterization of rhenium-doped MoS₂ single crystals. *J. Crystal Growth* **205**, 543–547 (1999).
- R5. R. J. Traill, A rhombohedral polytype of molybdenite. *Can. Mineral.* **7**, 524–526 (1962).
- R6. Y. Jing *et al.* Tuning electronic and optical properties of MoS₂ monolayer via molecular charge transfer. *J. Mater. Chem. A* **2**, 16892–16897 (2014).
- R7. D. Liu *et al.* Sulfur vacancies in monolayer MoS₂ and its electrical contacts. *Appl. Phys. Lett.* **103**, 183113 (2013).

- R8. J. Su *et al.* Tuning the electronic properties of bondings in monolayer MoS₂ through (Au, O) co-doping. *RSC Adv.* **5**, 68085–68091 (2015).
- R9. A. P. Nayak *et al.* Pressure-induced semiconducting to metallic transition in multilayered molybdenum disulphide. *Nat. Commun.* **5**, 3731 (2014).
- R10. R. M. A. Lieth and J. C. J. M. Terhell, Transition Metal Dichalcogenides in *Preparation and Crystal Growth of Materials with Layered Structures* edited by R. M. A. Lieth, Springer Science, 1977.
- R11. A. Togo and I. Tanaka, First principles phonon calculations in materials science. *Scr. Mater.* **108**, 1–5 (2015).
- R12. A. Shmeliov *et al.* Unusual stacking variations in liquid-phase exfoliated transition metal dichalcogenides. *ACS Nano* **8**, 3690–3699 (2014).
- R13. K. Dolui *et al.* Possible doping strategies for MoS₂ monolayers: An ab initio study. *Phys. Rev. B* **88**, 075420 (2013).
- R14. P. Zhao *et al.* Electronic and magnetic properties of Re-doped single-layer MoS₂: a DFT study. *Comput. Mater. Sci.* **128**, 287–293 (2017).
- R15. T. L. Tan, M.-F. Ng and G. Eda, Stable monolayer transition metal dichalcogenide ordered alloys with tunable electronic properties. *J. Phys. Chem. C* **120**, 2501–2508 (2016).

Reviewers' comments:

Reviewer #1 (Remarks to the Author):

The authors have taken serious consideration of reviewer's comments. However, I am still not satisfied with the investigation of the mechanism of the formation of 3R phase. I hope the authors consider to propose a possible mechanism for the formation mechanism, which will shine light to the future doping of 2D materials. Then the paper can be accepted for publication.

Reviewer #2 (Remarks to the Author):

Authors have made significant corrections/modifications in their manuscript. I recommend the acceptance of the manuscript once the following concerns are addressed

1) a) "Substitutional doping of traditional bulk semiconductors, the atomistic substitution with non-isoelectronic impurities, has been a core front-end process for semiconductor industry. Its primary purpose lies in defining the type of majority charge carriers and modulating their concentrations to a wide extent, such that the semiconductors can electrically functionalize as the key component in electronic and optoelectronic devices."

The beginning of the introductory paragraph should be restructured. Citation of more references in support of the statements, is essential.

b) "For the emerging two-dimensional (2D) or layered semiconductors such as transition metal dichalcogenides, MX_2 (M = transition metal and X = chalcogen), this conventional approach is still highly desired especially considering the issues specific to them: natively unipolar conduction¹, substantial contact resistance^{2,3}, and inferior stability and controllability of other available doping techniques, e.g., electrostatic gating⁴ and surface molecular doping⁵."

For example, meaning of the aforementioned lines, is not so clear. Besides, authors should include a few recent works, as mentioned below.

i) Lateral MoS₂ p–n Junction Formed by Chemical Doping for Use in High-Performance Optoelectronics, ACS Nano, 2014, 8 (9), pp 9332–9340

ii) Anisotropic transport in 1T' monolayer MoS₂ and its metal interfaces, Phys. Chem. Chem. Phys., 2017, 19, 10453-10461

2) Again if you cannot portray the intermediate stages of structural transformation, then saying "Nb-doping triggered 2H (ABAB) to 3R (ABCABC) structural transformation in bulk MoS₂ crystals" is misleading!! For the DFT studies, better you talk about 2H, 3R and % Nb-doped 3R (not structural transformation due to doping).

Authors said

"It implies that the 3R phase becomes increasingly energetically more favourable when Nb dopant is added to the supercell regardless of its size (The 6×6 (4×4) supercell is employed for \textcircled{A} and \textcircled{B} (\textcircled{C}) doping configurations). Furthermore, the dynamic stability of such 3R MoS₂:Nb bilayer is confirmed by the phonon dispersion showing no imaginary part (Supplementary Figure 4)."

a) What is meaning of red 2H bar (Figure 1 (b))? [For which the $E_{3R0}-E_{2H0}$ is positive!] Is this (per atom total energy of undoped 3R supercell) - (per atom total energy of 2H supercell)? Okay if

that has to be true, then total energy of 2H sample is more negative (compared to 3R). Perhaps, it should be (because 2H is reportedly the most stable structure)!! Now considering Nb-doping, how the 1.4%, 2.8% and 3.1% doped samples are showing negative $E_{3R0} - E_{2H0}$ (meV/atom)?? Did authors wanted to mean_

"We found that the 3R phase is energetically more stable than the 2H phase once Nb dopant is added to MoS₂, and it becomes increasingly more stable upon further Nb incorporation."

If so, then authors should indicate this in the main manuscript!

b) It's nice that the authors included the phonon dispersion diagram (of the Nb-doped 3R_MoS₂ supercell) in order to prove dynamical stability. However, for the sake of clarity and comprehensive understanding, authors should include the following studies which illustrate the dynamical stability of various poly types of MoS₂.

[1] Science 12 Dec 2014: Vol. 346, Issue 6215, pp. 1344-1347 DOI: 10.1126/science.1256815

[2] Appl. Phys. Lett. 108, 253106, 2016, doi: <http://dx.doi.org/10.1063/1.4954257>.

3)"Figure 2c shows the partial density of states (PDOS) plot for 2L undoped and Nb-doped MoS₂ in the 3R-type stacking order. Upon Nb doping, the indirect bandgap becomes narrower and the Fermi level (EF) downshifts converting the polarity of MoS₂ from n- to a p-type semiconductor, consistent with our experimental observation."

With the change of k-points, supercell size etc. the position of the EF had to change, and it happened (as we see the PDOS diagram of the 2L undoped MoS₂). Now it's closer to the C.B. edge. (which was previously nearer to the V.B. edge). While computing the DOS of any finite band gap material (in case MoS₂) using DFT, the position of the EF must be calibrated either at the V.B._max edge or, at the middle of the finite-gap. So, relating such numerical anomaly, with the intrinsic n-type nature of MoS₂ (experimentally observed) may not be appropriate!

Point-by-Point Responses to Reviewer #1's Comments

R1-1. *The authors have taken serious consideration of reviewer's comments. However, I am still not satisfied with the investigation of the mechanism of the formation of 3R phase. I hope the authors consider to propose a possible mechanism for the formation mechanism, which will shine light to the future doping of 2D materials. Then the paper can be accepted for publication.*

Response: First of all, we greatly appreciate the reviewer's positive comments on our revised manuscript. As being suggested by the reviewer, we would like to further consider the origin of the change in stacking of Nb-doped MoS₂. Indeed, it is the thermodynamic driving force that reorders the stacking stability of 3R over 2H as corroborated in our DFT calculations (Figure 1b), *i.e.* the 3R phase will naturally form once the Nb dopants are included in the crystal growth at elevated temperatures of ~1000 °C for >2 weeks. The relative energetic stability can be related to free-carrier screening by holes residing in the d_{z^2} bands with some delocalization [our Ref. #23: R. S. Title and M. W. Shafer, Phys. Rev. Lett. 28, 808–810 (1972)] in the Nb-doped MoS₂. It may lower the total energy of the system more effectively for non-centrosymmetric 3R-MoS₂ which originally tends to have more electric dipoles [M. Zhao et al. Light: Science & Applications 5, e16131 (2016) and E. Mishina et al. Appl. Phys. Lett. 106, 131901 (2015)]. We note the 3R phase was also experimentally observed in other heavily doped MoS₂ [our Ref. #27: K. K. Tiong et al. J. Crystal Growth 205, 543 (1999)] and theoretically suggested in defective MoS₂ [N. Cortés et al. arXiv:1710.00049v1 (2017)]. For clarification, we would also like to point out that we are *not* claiming that the 2H phase first crystalizes during the synthesis, and subsequently when the Nb dopants diffuse in, the 2H crystal transforms into the 3R crystal; hence the picture of phase transition with a middle stage is not necessarily required to visualize the process. However, such dynamic structural phase transition can be indeed potentially a follow-up study, given the fact that electrostatic gating [Y. Wang et al. Nature 550, 487–491 (2017)] or electron beam irradiation [A. Yan et al. J. Phys. Chem. C 121, 22559–22566 (2017)] are recently reported to induce the phase transition of 2D materials.

Relevant changes made:

- New sentences are added to further discuss the origin of structural transition (Page 4–5).

Point-by-Point Responses to Reviewer #2's Comments

Authors have made significant corrections/modifications in their manuscript. I recommend the acceptance of the manuscript once the following concerns are addressed.

Response: We thank the reviewer for her/his remarks and we are glad that the reviewer finds our work potentially suitable for publication in *Nature Communications*. Below we have addressed all remaining concerns in Introduction and DFT parts.

R2-1. *“Substitutional doping of traditional bulk semiconductors, the atomistic substitution with non-isoelectronic impurities, has been a core front-end process for semiconductor industry. Its primary purpose lies in defining the type of majority charge carriers and modulating their concentrations to a wide extent, such that the semiconductors can electrically functionalize as the key component in electronic and optoelectronic devices.”*

The beginning of the introductory paragraph should be restructured. Citation of more references in support of the statements, is essential.

“For the emerging two-dimensional (2D) or layered semiconductors such as transition metal dichalcogenides, MX_2 (M = transition metal and X = chalcogen), this conventional approach is still highly desired especially considering the issues specific to them: natively unipolar conduction¹, substantial contact resistance^{2,3}, and inferior stability and controllability of other available doping techniques, e.g., electrostatic gating⁴ and surface molecular doping⁵.”

For example, meaning of the aforementioned lines, is not so clear. Besides, authors should include a few recent works, as mentioned below.

i) Lateral MoS_2 p–n Junction Formed by Chemical Doping for Use in High-Performance Optoelectronics, ACS Nano, 2014, 8 (9), pp 9332–9340

ii) Anisotropic transport in 1T' monolayer MoS_2 and its metal interfaces, Phys. Chem. Chem. Phys., 2017, 19, 10453-10461.

Response: Taking the reviewer’s suggestions seriously, the opening sentences are reconstructed to deliver a clearer message about general aspects of substitutional doping of semiconductors. We also comprehensively describe the importance and current stages of substitutional doping for 2D materials. In this context, the suggested recent articles (surface molecular doping and 2H-1T' phase transition) are also cited in the revised manuscript.

Relevant changes made:

- First 4 sentences are completely modified in the beginning of introduction (Page 3, 1st paragraph).
- The suggested two references by the referee are added in an introduction.

R2-2. *Again if you cannot portray the intermediate stages of structural transformation, then saying “Nb-doping triggered 2H (ABAB) to 3R (ABCABC) structural transformation in bulk MoS_2 crystals” is misleading!! For the DFT studies, better you talk about 2H, 3R and % Nb-doped 3R (not structural transformation due to doping). Authors said “It implies that the 3R phase becomes increasingly energetically more favourable when Nb dopant is added to the supercell regardless of its size (The 6×6 (4×4) supercell is employed for \textcircled{A} and \textcircled{B} (\textcircled{C}) doping configurations). Furthermore, the dynamic stability of such 3R MoS_2 :Nb bilayer is confirmed by the phonon dispersion showing no imaginary part (Supplementary Figure 4).”*

a) What is meaning of red 2H bar (Figure 1 (b))? [For which the $E_{3R0}-E_{2H0}$ is positive!] Is this (per atom total energy of undoped 3R supercell) - (per atom total energy of 2H supercell)? Okay if that has to be true, then total energy of 2H sample is more negative (compared to 3R). Perhaps, it should be (because 2H is reportedly the most stable structure)!! Now considering Nb-doping, how the 1.4%, 2.8% and 3.1% doped samples are showing negative $E_{3R0}-E_{2H0}$ (meV/atom)?? Did authors wanted to mean “We found that the 3R phase is energetically more stable than the 2H phase once Nb dopant is added to MoS_2 , and it becomes increasingly more stable upon further Nb incorporation.” If so, then authors should indicate this in the main manuscript!

b) It’s nice that the authors included the phonon dispersion diagram (of the Nb-doped 3R MoS_2 supercell) in order to prove dynamical stability. However, for the sake of clarity and comprehensive understanding, authors should include the following studies which illustrate the dynamical stability of various poly types of MoS_2 .

[1] *Science* 12 Dec 2014: Vol. 346, Issue 6215, pp. 1344-1347 DOI: 10.1126/science.1256815

[2] *Appl. Phys. Lett.* 108, 253106, 2016, doi: <http://dx.doi.org/10.1063/1.4954257>.

Response: We agree with the reviewer, so we modified the titles of Figure 1 and the first subsection of results as well as explanation for DFT calculations to prevent any misleading about the potential role of Nb dopants in terms of structural transformation. As for the point (a), we clarify that the energy difference is indeed as stated: (per atom total energy of undoped 3R supercell) - (per atom total energy of 2H supercell), *i.e.*, it is the energy of the 3R structure with reference to the 2H structure. Hence if the 3R phase is less stable than the 2H phase, this value has to be positive and vice versa. In fact, our corresponding DFT calculations, Figure 1(b), can be interpreted as mentioned by the reviewer. For further clarification, we changed its figure caption to: "Calculated energy difference of 3R phase with respect to 2H phase, with positive (negative) values indicating that the 2H (3R) phase is more stable". We also change the descriptive statement in the main text following the reviewer's suggestion: "We note that the 3R phase is energetically more stable than the 2H phase once the Nb dopant is added to the MoS₂ supercell. And, it becomes increasingly more stable upon further Nb incorporation". The labelling in Figure 1(b) is modified to avoid confusion as well. Lastly, we agree on the reviewer's point (b), thus now including those two representative works in the discussion of dynamic stability of bilayer 3R Nb-doped MoS₂.

Relevant changes made:

- First subsection of results is now titled as "Crystal structure of Nb-doped MoS₂".
- Title of Figure 1 is modified to "Crystal structure of undoped and Nb-doped bulk MoS₂".
- "Doping-induced structural transformation" is deleted when explaining a motivation of DFT calculations (Page 4, 2nd paragraph).
- Descriptive sentences for the DFT calculations are rephrased in a clear way following the reviewer's suggestion (Page 4, 2nd paragraph).
- Figure caption and label (Figure 1b) are modified for further clarification.
- Two references, mentioned by the referee, are added in the main text along with a corresponding phrase (Page 4, 2nd paragraph).

R2-3. "Figure 2c shows the partial density of states (PDOS) plot for 2L undoped and Nb-doped MoS₂ in the 3R-type stacking order. Upon Nb doping, the indirect bandgap becomes narrower and the Fermi level (E_F) downshifts converting the polarity of MoS₂ from *n*- to a *p*-type semiconductor, consistent with our experimental observation."

*With the change of k -points, supercell size etc. the position of the E_F had to change, and it happened (as we see the PDOS diagram of the 2L undoped MoS₂). Now it's closer to the C.B. edge. (which was previously nearer to the V.B. edge). While computing the DOS of any finite band gap material (in case MoS₂) using DFT, the position of the E_F must be calibrated either at the V.B. _max edge or, at the middle of the finite-gap. So, relating such numerical anomaly, with the intrinsic *n*-type nature of MoS₂ (experimentally observed) may not be appropriate!*

Response: We agree with reviewer, and the Fermi level was simply read off from DFT calculations. Thus, we have deleted a relevant comment of intrinsic *n*-type of undoped MoS₂, and the PDOS figure is updated with calibrated E_F to the middle of bandgap for undoped MoS₂ (Figure R2-1).

Relevant changes made:

- Main Figure 2(c) is updated with the calibrated E_F as shown in Figure R2-1.
- Corresponding sentence for the DFT calculations (PDOS) is modified taking the reviewer's suggestion (Page 6, 3rd paragraph).

Figure R2-1. Calculated partial density of states (PDOS) of 2L undoped and Nb-doped MoS₂ (3R-stacked) projected on the selected atoms, Mo, Nb and S. For the intrinsic MoS₂, E_F is calibrated at the middle of the bandgap. For the bilayer MoS₂:Nb, two representative Mo atoms, the nearest to and remote away from Nb dopant, are shown here considering their distinct contribution to the valence band maximum.

REVIEWERS' COMMENTS:

Reviewer #1 (Remarks to the Author):

I will happy for the paper to be accepted for publication.

Reviewer #2 (Remarks to the Author):

Revised version could be accepted for publication provided authors remove the following statement:

"This is due to their inferior stability and controllability compared to that of traditional substitutional doping which is secured with covalent bonding"

which is unnecessary and might be incorrect. Research groups around the world are fabricating phase-engineered MoS₂ based devices and they appear to be pretty stable.

Point-by-Point Responses to Reviewer #1's Comments

I will happy for the paper to be accepted for publication.

Response: We thank the referee for the kind comments.

Point-by-Point Responses to Reviewer #2's Comments

Revised version could be accepted for publication provided authors remove the following statement: "This is due to their inferior stability and controllability compared to that of traditional substitutional doping which is secured with covalent bonding" which is unnecessary and might be incorrect. Research groups around the world are fabricating phase-engineered MoS₂ based devices and they appear to be pretty stable.

Response: We thank the reviewer for her/his remarks and we have deleted the corresponding sentence upon suggestion.